# CROSS-MODALITY MASKED PRE-TRAINING FOR VISIBLE-INFRARED PERSON RE-IDENTIFICATION

## ABSTRACT

Visible-Infrared person re-identification is a challenging yet important task in the field of intelligent surveillance. Most existing approaches focus on designing powerful deep networks to learn modality-shared representations, while little attention has been paid to using pre-training methods, although they can improve the performance of cross-modality tasks stably. This paper proposes a cross-modality masked pre-training (CMMP) method for visible-infrared person re-identification. Specifically, we generate color-irrelevant images using random channel exchangeable augmentation to minimize the difference between modalities at first. In the pre-training process, the visible together with the generated image, and the infrared image are masked by sharing the same random mask. Considering the misalignment of visible and infrared images in the datasets, we then reconstruct the masked areas only of the visible and the generated images using a lightweight decoder, which makes the pre-training process more efficient. Extensive experiments on two visible-infrared person re-identification datasets verify the effectiveness of the proposed method. CMMP outperforms the baseline method by +1.87% and +1.24% mAP on SYSU-MM01 and RegDB, respectively.

## 1 INTRODUCTION

Person re-identification (ReID), which aims to retrieve a specific person in non-overlapping camera networks, has received much attention in recent years. Numerous prior studies Wu et al. (2016); Sun et al. (2018); He et al. (2021); Ma et al. (2023) have dedicated their efforts to the realm of Person ReID using RGB images, while only a few researchesWang et al. (2019b); Zhang et al. (2019); Li et al. (2020); Duan et al. (2020) take infrared images into consideration, that is, visible-infrared Person ReID (VI-ReID).

Most of the VI-ReID approaches primarily concentrate on minimizing the dissimilarities between modalities during the training phase. Interestingly, there is a limited body of research that specifically delves into cross-modality pre-training. In contrast to the conventional practice of employing models pre-trained on ImageNetDeng et al. (2009) directly for downstream tasks in cross-modality Person ReID, the specialized cross-modality pre-training methods not only address potential issues arising from modality misalignment between pre-training and training data but also facilitates information exchange between modalities during the pre-training phase. This dual-pronged approach serves the purpose of diminishing the discrepancies between modalities, thereby simplifying the process of cross-modality matching during model fine-tuning. Fig. 1 illustrates this distinction.

In recent times, self-supervised learning techniques have garnered remarkable success within the realm of computer vision. Among these, methods such as MAEHe et al. (2022) and SimMIMXie et al. (2022), which rely on the mask reconstruction strategy, have achieved noteworthy milestones in the domain of self-supervised pre-training. These approaches involve the utilization of asymmetric encoders and decoders, where only the unmasked image portion is passed to the encoders, and subsequently, the masked section is reconstructed through the decoders. This methodology effectively harnesses the model's expressive capacity, thereby furnishing an exceptional initialization for the fine-tuning of downstream tasks.

Building upon the significant achievements in self-supervised learning, we introduce a novel cross-modality self-supervised pre-training method tailored for the VI-ReID task, namely Cross-Modality

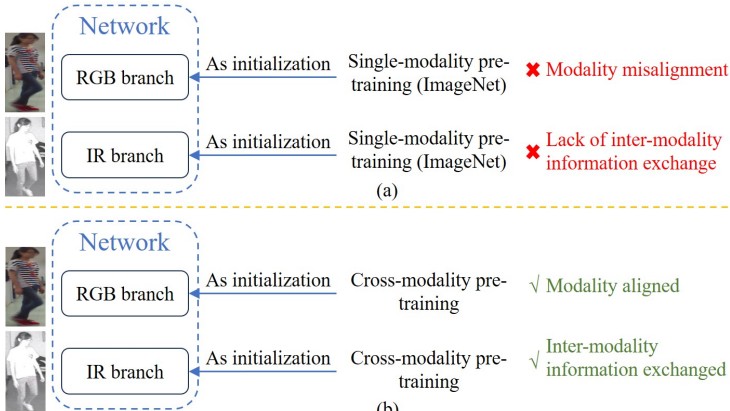

Figure 1: The difference between general pre-training methods and specially designed cross-modality pre-training methods in cross-modality person ReID. (a) and (b) are the general pre-training methods and the cross-modality methods, respectively.

Masked Pre-training (CMMP). CMMP is designed to concurrently learn from both visible and infrared images, thereby enhancing feature representation and offering robust support for fine-tuning in VI-ReID tasks. The CMMP method unfolds in the following manner: First, the visible image undergoes a random channel exchange augmentation, where a random channel is substituted with data from other channels. This process yields a generated image with a visual appearance closely resembling that of an infrared image. Subsequently, the visible image, the randomly channel-exchanged generated image, and the infrared image are collectively processed through a shared random mask. These three images are then subjected to masking and passed through a CNN encoder, allowing for the joint modeling of characteristics from both visible and infrared images. Given the distinct characteristics of the VI-ReID datasets, achieving a one-to-one alignment between visible and infrared images can be challenging. Therefore, in the decoder section, the reconstruction is limited to the mask segment of the visible image and the generated image, excluding the infrared image. The encoder aligns with the training framework utilized for downstream VI-ReID tasks, while the decoder is extremely lightweight, housing only a single linear layer. Experimental results underscore the efficacy of our proposed cross-modality mask pre-training method in the simultaneous learning of visible and infrared images, thereby yielding robust feature expressions. Upon fine-tuning in the context of VI-ReID, the model's performance sees a substantial improvement, surpassing the benchmarks set by existing state-of-the-art methods.

To summarize, our contributions are summarized as follows:

- We introduce the mask reconstruction paradigm into the field of VI-ReID for the first time, and achieve significant performance improvements.
- We implement a MAE-like architecture based on convolutional neural networks and verify its effectiveness.
- We achieve remarkable performance results that surpass existing advanced methods on two VI-ReID datasets.

## 2 RELATED WORK

### 2.1 CROSS-MODALITY PRE-TRAINING

Cross-modality pre-training is a deep learning technique that facilitates the learning of representation relationships between data originating from various modalities, which can encompass images, text, audio, and more. It has made great progress in the intersection of language and vision. ViLBERLu et al. (2019) is a pioneer in this field, which learns the association between vision and language through joint pre-training. LXMERTTan & Bansal (2019) and UniVLLuo et al. (2020) adopt the

transformer architecture and learn cross-modality encoder representations by jointly training image-text and video-text data.

Similarly, some researchers have applied cross-modality pre-training methods in other fields, such as audio-text intersection fields. UniSpeechWang et al. (2021) proposes a unified speech representation learning method, which can be applied to a variety of speech-related tasks, such as speech recognition, speech synthesis and speech emotion recognition, etc. In the field of intersection of RGB and depth images, Yan et al.Yan et al. (2022) proposes a multi-modal mask pre-training method M$^3$PT, which first shares a random mask to mask the panoramic RGB image and sparse depth map, and then masks the mask area. Sparse depth reconstruction.

## 2.2 VISIBLE-INFRARED PERSON REID

The visible-infrared person ReID (VI-ReID) problem is first raised by Wu et al.Wu et al. (2017) in 2017. In addition to proposing a cross-modality person ReID framework, they also provide a public large-scale VI-ReID dataset, which is known as the widely used SYSU-MM01. After that, a large amount of research work poured into this field. In 2018, Dai et al.Dai et al. (2018) applies the idea of generative adversarial networks to VI-ReID, and designs a discriminator based on cutting-edge generative adversarial training to solve the problem of insufficient discriminative information to learn discriminative feature representations from different modalities. Hi-CMDChoi et al. (2020) can automatically extract ID discriminating factors and ID-independent factors from visible and infrared images, and then use the ID discriminating factors for robust cross-modality matching. Ye et al.Ye et al. (2021) randomly exchanges color channels to uniformly generate color-independent images, which can be seamlessly integrated into existing enhancement operations without modifying the network, thus continuously improving the robustness to color changes.

## 3 METHOD

In this section, we introduce the Cross-Modality Masked Pretraining (CMMP) framework. CMMP is a method similar to MAEHe et al. (2022) and SimMIMXie et al. (2022), built upon Convolutional Neural Networks (CNNs). Like these methods, CMMP comprises an encoder responsible for mapping the observed signal into a latent representation, and a decoder that reconstructs the original signal using this latent representation. The encoder operates on sections of the observed signal, both with and without the masked portion, and employs a lightweight decoder to reconstruct the complete signal using the latent representation and the masked tokens. In contrast to single-modality masked pre-training, CMMP takes three images as input: the visible image, a generated image resulting from random channel exchange (following the approach in Ye et al. (2021)), and the infrared image. Due to the misalignment inherent in the VI-ReID dataset, CMMP focuses its reconstruction efforts solely on the visible image and the generated images after masking. Fig. 2 provides a detailed illustration of the CMMP framework for VI-ReID, consisting of four main components:

**Masking Strategy.** Given visible and infrared images of the same person, and an image generated by random channel exchange of the visible image, this component designs how to select the area to be masked, and how to implement the masking of the selected area. The transformed image after masking will be used as input to the encoder.

**Encoder Architecture.** The encoder is used to extract the latent feature representations of the masked images, and then these latent feature representations are used as the input of the decoder. In this method, the encoder is a CNN architecture.

**Decoder Architecture.** The latent feature representations are input into the decoder to reconstruct the original signal of the masked region.

**Prediction Target.** This component defines the form of the raw signal to be predicted. It can be the original pixel value or the conversion of the original pixel. In this method, the prediction target is the original pixel value. This component also defines the loss function, which typically includes cross-entropy and $l_1, l_2$ losses, etc.

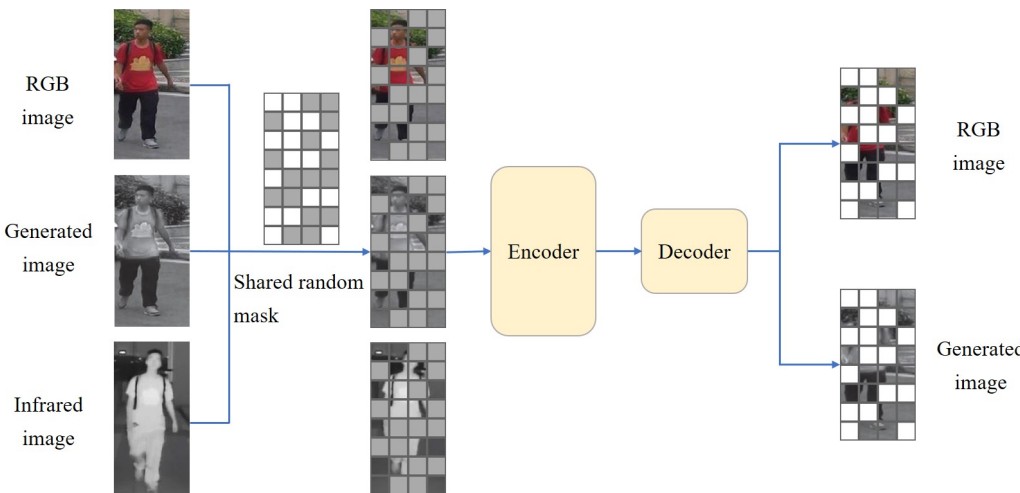

Figure 2: Illustration of the cross-modality masked pre-training (CMMP) framework for VI-ReID.

## 3.1 MASKING STRATEGY

Our masking strategy differs from that of MAEHe et al. (2022) and SimMIMXie et al. (2022), which focus on masking single-modality visible data. In our proposed CMMP method, we employ shared random masks to simultaneously mask visible images, generated images resulting from random channel exchange, and infrared images, thus generating masked image pairs for pre-training. Both MAE and SimMIM employ the transformer architecture to divide images into equally sized image blocks and perform masking operations. In their approaches, an image block is either entirely visible or fully obscured, with the masked part being completely discarded. In contrast, CMMP leverages the CNN architecture and retains the masked area, aligning with practices adopted by the NLP community [devlin2018bert, liu2019roberta] and SimMIM. In CMMP, each mask patch is substituted with a learnable mask token vector, and the entire image, including the mask tokens, is fed into the encoder. The dimensions of the mask tokens match those of other areas within the image.

For mask region selection, we to randomly sample image patches according to a uniform distribution without replacement, since uniform distribution prevents potential center bias. We set the mask block to a square size with an aspect ratio of 1:1 like MAEHe et al. (2022) and SimMIMXie et al. (2022) .The mask rate is set to 0.8. Random sampling with a high mask rate eliminates redundancy to a large extent and increases the difficulty of the reconstruction task so that it cannot be easily accomplished by extrapolation from visible adjacent image patches (such as linear interpolation).

## 3.2 ENCODER ARCHITECTURE

Different from MAEHe et al. (2022) and SimMIMXie et al. (2022) which use transformer architectures as the encoder, our CMMP is based on CNN architecture. The encoder selected in this section uses the Channel Augmented Joint Learning (CAJL) methodYe et al. (2021). Its backbone network is a non-local attention module with ResNet50He et al. (2016), of which the architecture is shown in Fig. 3.

## 3.3 DECODER ARCHITECTURE

The main goal of the decoder is to complete the reconstruction of the mask part by predicting the pixel values of the mask part. In principle, the decoder can be of any shape as long as it can correctly process the input signal and complete the data reconstruction goal. In this section, unlike MAEHe et al. (2022), which uses a relatively heavyweight network, such as ViTDosovitskiy et al. (2020), as the decoder, CMMP decoder consists of only one linear layer to predict the pixel values of the mask part.

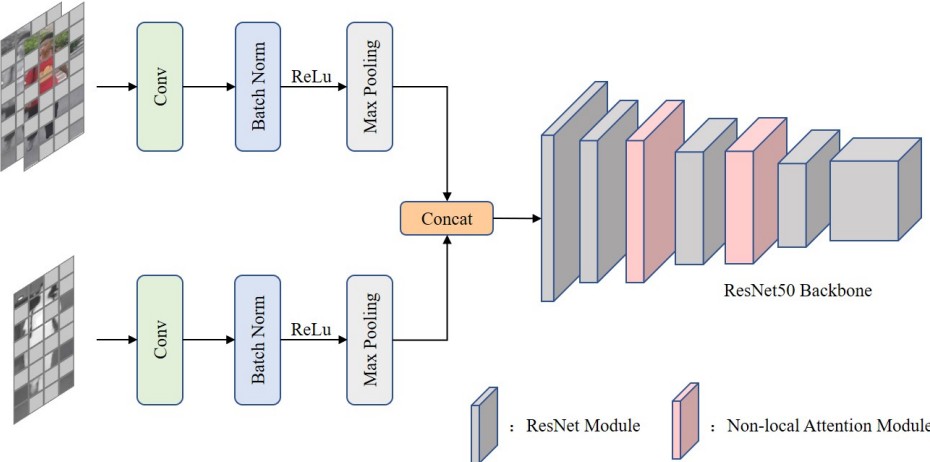

Figure 3: The encoder architecture of CMMP.

## 3.4 PREDICTION TARGET

To accomplish the mask reconstruction task, CMMP takes pixel values as its prediction target. Specifically, after the masked visible image, random channel exchange generated image, and infrared image are encoded by the encoder, the decoder maps the features from the encoder back to the original resolution to predict all pixel values of the image. Finally, the loss is used to guide the reconstruction of visible images and generated images:

$$\mathcal{L} = \frac{1}{\Omega(X_M)} \|Y_M - X_M\| \tag{1}$$

where $X, Y \in \mathbb{R}^{3HW}$ are the input masked visible image and the random channel exchange generated images, and the reconstructed visible and generated images; $M$ represents the masked pixel part; $\Omega(\dot{)}$ represents the number of masked pixels.

## 4 EXPERIMENTS

### 4.1 DATASETS

To evaluate the effectiveness of CMMP, we carry out experiments on two widely-used VI-ReID benchmarks: SYSU-MM01Wu et al. (2017) and RegDBNguyen et al. (2017). SYSU-MM01Wu et al. (2017) collects images from four visible and two infrared cameras in indoor and outdoor environments. The training set contains 395 different identities, including 22,258 visible and 11,909 infrared images. The test set contains 96 different identities, including 3,803 infrared images as query images, and 301 visible and 3,010 visible images as gallery for single-shot and multi-shot, respectively. RegDBNguyen et al. (2017) contains 412 identities and 8,240 images, each identity has 10 different infrared images and 10 different visible images. Among them, 206 identities are used for training and 206 identities are used for testing. This dataset also provides different dataset partitions for 10 experiments, allowing the model to be evaluated for 10 experiments to obtain statistically stable results.

### 4.2 IMPLEMENTATION DETAILS

During the pre-training process, we first convert the image to $288 \times 144$ size, and then use random cropping and random horizontal flipping for image augmentation. We use the Stochastic Gradient Descent (SGD) optimizer for optimization and train for 400 epochs using a distributed training method on a workstation of 4 Nvidia Titan Xp GPUs with 12 GB VRAM. The initial learning rate is set to 0.1 and decays by 0.1 and 0.01 at 20 and 50 epochs respectively. The training batch size is set to 128, evenly distributed across 4 GPUs. During fine-tuning, the optimizer selection and

learning rate settings are consistent with the pre-training stage, but only train for 100 epochs on a single Nvidia Titan Xp GPU. The training batch size is set to 8, which includes 4 visible images and 4 infrared images.

### 4.3 ABLATION STUDY

In this section, we conduct ablation study on masking strategy, prediction target and pre-training epochs. All experiments are carried out on SYSU-MM01Wu et al. (2017).

#### 4.3.1 MASKING STRATEGY

**Masking Size.** We employ square-shaped masks with consistent width and height to mask images. We conduct experiments using multiple sets of masks of different sizes with mask rates set at 0.75 and 0.8, respectively. Table 1 illustrates the impact of different mask sizes on the final results, where the values for mask size represent the number of pixels occupied by the side length of the mask block.

| Mask Ratio | 0.75 | | | 0.8 | | |
|---|---|---|---|---|---|---|
| Mask Size | mAP | R1 | R5 | mAP | R1 | R5 |
| 8 | 12.89 | 10.54 | 34.21 | 15.48 | 15.09 | 39.89 |
| 12 | 26.23 | 27.66 | 55.54 | 12.21 | 11.10 | 31.61 |
| 16 | **67.31** | 70.42 | 89.88 | **67.49** | **71.97** | **90.88** |
| 18 | 65.07 | 68.60 | 90.22 | 65.46 | 69.92 | 90.17 |
| 24 | 66.61 | **71.18** | **90.72** | 26.21 | 24.66 | 55.40 |
| 36 | 29.70 | 29.92 | 60.56 | 22.00 | 20.62 | 49.70 |

Table 1: The impact of different mask sizes on the final results at mask ratios of 0.75 and 0.8. Best results are markes in bold.

From Table 1, it can be observed that regardless of whether the mask rate is 0.75 or 0.8, the mAP is highest when the mask size is $16 \times 16$. Furthermore, when evaluating R1 and R5, the $16 \times 16$ mask block still performs the best at a mask rate of 0.8. However, at the mask rate of 0.75, the $24 \times 24$ mask block surpasses the $16 \times 16$ block in both R1 and R5, reaching the highest values but exhibits a sharp decline at a mask rate of 0.8.

Overall, the optimal mask size for fine-tuning pre-trained models on downstream tasks has been determined to be the $16 \times 16$ mask block. Consequently, this mask scale is selected for all subsequent experiments. Notably, when the mask size deviates either towards being too large or too small, it leads to a significant degradation in the final results. When the mask size is too small, individual mask regions become excessively petite, rendering it easier for the model to recover information from the surrounding areas. This, however, hinders the model from acquiring discriminative feature representations, and the inflow of cross-modality information from different axes further diminishes performance during fine-tuning. Conversely, when the mask size is too large, each mask region lacks a substantial amount of information, making it challenging to establish dependencies between visible and infrared images. This also has an adverse impact on the model's overall performance capabilities.

**Masking Ratio.** Building upon the experiments conducted to select the mask size, this section focuses on mask ratio selection, with the use of $16 \times 16$ mask blocks as the baseline. Table 2 provides a clear overview of the results obtained for different mask ratios. It is evident that a mask ratio of 0.8 yields the best fine-tuning results for the model, surpassing the performance of other mask ratios in terms of both mAP and R1 accuracy. Furthermore, as the mask ratio gradually increases from 0.5 to 0.8, the fine-tuning results of the model exhibit improvement. This trend suggests that a relatively high mask ratio serves to obscure more image regions, rendering it challenging for the model to simply infer the reconstruction task from adjacent unmasked image portions. Consequently, this increased difficulty in the reconstruction task enables the model to learn more discriminative features. However, it's important to note that when the mask ratio reaches 0.85, the model's mAP during fine-tuning experiences a 2.44% decline compared to the optimal result obtained at a mask ratio of 0.8. This observation underscores that an excessively high mask ratio obscures too many image regions,

leading to substantial information loss that hampers the successful completion of the reconstruction task, ultimately diminishing the model's performance. In conclusion, the selection of an appropriate mask ratio during mask reconstruction is crucial, as both excessively high and excessively low mask ratios can adversely impact the model's performance. Therefore, the choice of mask ratio should be tailored to the specific task at hand to effectively improve model performance.

| Mask Ratio | mAP | R1 | R5 |
|:---:|:---:|:---:|:---:|
| 0.5 | 30.33 | 32.60 | 62.14 |
| 0.55 | 36.75 | 40.26 | 71.13 |
| 0.6 | 64.74 | 69.66 | 90.17 |
| 0.65 | 65.68 | 70.52 | 90.59 |
| 0.7 | 67.35 | 71.05 | 90.80 |
| 0.75 | 67.31 | 70.42 | 89.88 |
| 0.8 | **67.49** | **71.97** | **90.88** |
| 0.85 | 65.05 | 68.95 | 90.01 |

Table 2: The impact of different mask ratios on the final results at mask size of $16 \times 16$. Best results are marked in bold.

### 4.3.2 PREDICTION TARGET

| Prediction Target | mAP | R1 | R5 |
|:---:|:---:|:---:|:---:|
| Visible image | 66.12 | 65.52 | 89.72 |
| Generated image | 66.29 | 69.92 | **91.72** |
| Infrared image | 63.80 | 65.97 | 89.11 |
| Visible + Generated images | **67.49** | **71.97** | 90.88 |
| Visible + Generated + Infrared images | 26.68 | 25.95 | 56.03 |

Table 3: The impact of different prediction targets on the final results. Best results are marked in bold.

Table 3 presents the impact of different prediction targets on the final results when using a mask size of $16 \times 16$ and a mask ratio of 0.8. In addition to separately predicting three types of images, we also considered simultaneously predicting visible and random channel exchange generated images, as well as predicting all three cross-modality images simultaneously.

We observed that simultaneously predicting both visible and generated images leads to the best fine-tuning results for the model. Specifically, it outperforms the second-best approach (predicting generated images alone) by a margin of 1.20% in terms of mAP. This indicates that the use of random channel exchange generated images, which closely resemble infrared images while retaining the original texture structure of visible images, helps reduce the disparities between modalities. This reduction in differences allows the model to better grasp cross-modality fusion information, ultimately enhancing performance during fine-tuning. However, when attempting to predict all three cross-modality images simultaneously, the model's performance is notably worse, considerably lower than when predicting other targets. This suggests that the misalignment between infrared images and visible/generated images has an adverse impact on prediction results, leading to a significant decrease in model performance during fine-tuning. Furthermore, among the results obtained when separately predicting the three types of images, the model performs best when predicting generated images alone. This underscores the valuable role played by random channel exchange generated images in reducing modality differences, facilitating feature learning, and supporting model transferability. As a result, in subsequent experiments, we opt to predict both visible and generated images.

**Predicting Masks vs. Predicting the Whole Image.** We also compare the difference between separately reconstructing invisible regions (predicting masks) and simultaneously reconstructing both invisible and visible regions (predicting the whole image). Both of these prediction methods involve learning from the restoration of the original signal image, but they vary in terms of their reconstruction targets. Table 4 presents the comparative results of these two prediction methods,

both using a mask size of $16 \times 16$ and a mask rate of 0.8, with the prediction targets being visible and random channel exchange generated images.

| Prediction Area | mAP | R1 | R5 |
|---|---|---|---|
| Mask | **67.49** | **71.97** | **90.88** |
| Whole image | 32.10 | 33.58 | 64.79 |

Table 4: The impact of different prediction areas on the final results. Best results are marked in bold.

The table clearly demonstrates that the strategy of predicting mask regions yields notably superior results compared to predicting all image pixels. Surprisingly, predicting all image pixels actually hampers the overall pre-training objective, leading to a decline in the model's performance during fine-tuning. This suggests that predicting mask regions is a more effective approach for representation learning.

### 4.3.3 PRE-TRAINING EPOCHS

| Pre-training Epochs | mAP | R1 | R5 |
|---|---|---|---|
| 100 | 67.36 | **72.07** | **91.27** |
| 200 | 66.69 | 69.31 | 90.59 |
| 300 | 67.45 | 71.68 | **91.27** |
| 400 | **67.49** | 71.97 | 90.88 |

Table 5: The impact of different pre-training epochs on the final results. Best results are marked in bold.

Table 5 provides an overview of how different pre-training epochs impact the model's performance during fine-tuning. In general, there isn't a substantial difference in accuracy when varying the number of pre-training epochs. However, as the number of pre-training epochs increases, the model's performance during fine-tuning shows improvement, with the highest mAP achieved at 400 epochs. This aligns with the conventional wisdom that more training epochs allow the model to acquire better feature representations, consequently enhancing its performance during fine-tuning. This improvement is further evidenced by the reduction in model loss as the number of epochs increases.

## 5 COMPARISON WITH SOTA METHODS

| Settings | All Srearch | | | | Indoor Search | | | |
|---|---|---|---|---|---|---|---|---|
| Method | R1 | R10 | R20 | mAP | R1 | R10 | R20 | mAP |
| Zero-PadWu et al. (2017) | 14.80 | 54.12 | 71.33 | 15.95 | 20.58 | 68.38 | 85.79 | 26.92 |
| HCMLYe et al. (2018) | 14.32 | 53.16 | 69.17 | 16.16 | 24.52 | 73.25 | 86.73 | 30.08 |
| eBDTRYe et al. (2019) | 27.82 | 67.34 | 81.34 | 28.42 | 32.46 | 77.42 | 89.62 | 42.46 |
| HSMEHao et al. (2019) | 20.68 | 32.74 | 77.95 | 23.12 | - | - | - | - |
| $D^2$RLWang et al. (2019b) | 28.9 | 70.6 | 82.4 | 29.2 | - | - | - | - |
| AlignGANWang et al. (2019a) | 42.4 | 85.0 | 93.7 | 40.7 | 45.9 | 87.6 | 94.4 | 54.3 |
| X-ModalLi et al. (2020) | 49.9 | 89.8 | 96.0 | 50.7 | - | - | - | - |
| Hi-CMDChoi et al. (2020) | 34.9 | 77.6 | - | 35.9 | - | - | - | - |
| cm-SSFTLu et al. (2020) | 47.7 | - | - | 54.1 | - | - | - | - |
| DDAGYe et al. (2020a) | 54.75 | 90.39 | 95.81 | 53.02 | 61.02 | 94.06 | 98.41 | 67.98 |
| HATYe et al. (2020b) | 55.29 | 92.14 | 97.36 | 53.89 | 62.10 | 95.75 | 99.20 | 69.37 |
| MCLNetHao et al. (2021) | 65.40 | 93.33 | 97.14 | 61.98 | 72.56 | 96.98 | 99.20 | 76.58 |
| CAJLYe et al. (2021) | 69.00 | 94.85 | 98.16 | 65.62 | 71.15 | **97.15** | **99.32** | 75.99 |
| CMMP | **71.97** | **95.53** | **98.66** | **67.49** | **75.23** | 96.51 | 99.18 | **78.28** |

Table 6: Comparison with SOTA methods on SYSU-MM01Wu et al. (2017). Best results are marked in bold.

| Settings | Visible to Infrared | | | | Infrared to Visible | | | |
|---|---|---|---|---|---|---|---|---|
| Method | R1 | R10 | R20 | mAP | R1 | R10 | R20 | mAP |
| Zero-PadWu et al. (2017) | 17.75 | 34.21 | 44.35 | 18.90 | 16.63 | 34.68 | 44.25 | 17.82 |
| HCMLYe et al. (2018) | 24.44 | 47.53 | 56.78 | 20.08 | 21.70 | 45.02 | 55.58 | 22.24 |
| eBDTRYe et al. (2019) | 34.62 | 58.96 | 68.72 | 33.46 | 34.21 | 58.74 | 68.64 | 32.49 |
| HSMEHao et al. (2019) | 50.85 | 73.36 | 84.66 | 47.00 | 50.15 | 72.40 | 81.07 | 46.16 |
| $D^2$RLWang et al. (2019b) | 43.4 | 66.1 | 76.3 | 44.1 | - | - | - | - |
| AlignGANWang et al. (2019a) | 57.9 | - | - | 53.6 | 56.3 | - | - | 53.4 |
| X-ModalLi et al. (2020) | 62.21 | 83.13 | 91.72 | 60.18 | - | - | - | - |
| Hi-CMDChoi et al. (2020) | 70.93 | 86.39 | - | 66.04 | - | - | - | - |
| cm-SSFTLu et al. (2020) | 72.3 | - | - | 72.9 | 71.0 | - | - | 71.7 |
| DDAGYe et al. (2020a) | 69.34 | 86.19 | 91.49 | 63.46 | 68.06 | 85.15 | 90.31 | 61.80 |
| HATYe et al. (2020b) | 71.83 | 87.16 | 92.16 | 67.56 | 70.02 | 86.45 | 91.61 | 66.30 |
| MCLNetHao et al. (2021) | 80.31 | 92.70 | 96.03 | 73.07 | 75.93 | 90.93 | 94.59 | 69.49 |
| CAJLYe et al. (2021) | 82.84 | 93.59 | **96.38** | 74.23 | 80.43 | **92.69** | 95.66 | 72.14 |
| CMMP | **83.37** | **93.87** | 96.31 | **75.47** | **80.64** | 92.56 | **95.72** | **72.68** |

Table 7: Comparison with SOTA methods on RegDBNguyen et al. (2017). Best results are marked in bold.

Table 6 and Table 7 show the results of different evaluation modes on SYSU-MM01Wu et al. (2017) and RegDBNguyen et al. (2017). The results demonstrate that our proposed CMMP outperforms most existing state-of-the-art methods. On SYSU-MM01Wu et al. (2017), CMMP outperforms the best-performing method, MCLNetHao et al. (2021) (except the baseline method CAJLYe et al. (2021)), by 5.51% and 1.70% in mAP for all and indoor searches, respectively, and also outperforms MCLNet by 6.57% and 2.67% in Rank-1 accuracy. Compared to the baseline method CAJL[194], CMMP significantly improves performance. In all and indoor search modes, CMMP achieves a 1.87% and 2.29% increase in mAP, and a 2.97% and 4.08% increase in Rank-1 accuracy, respectively. These results emphasize the effectiveness of CMMP.

On RegDBNguyen et al. (2017), CMMP also achieves the best performance. It outperforms other existing SOTA methods in both visible to infrared and infrared to visible searching settings. In terms of mAP, CMMP improves performance by 2.40% and 3.19% compared to the best-performing method MCLNetHao et al. (2021) (except the baseline method CAJLYe et al. (2021)), and it also improves Rank-1 accuracy by 3.06% and 4.71% in the two searching modes, respectively. Compared to the baseline method CAJLYe et al. (2021), CMMP performs well in both evaluation modes on the RegDB dataset, with an increase of 1.24%/0.54% in mAP and 0.53%/0.21% in Rank-1 accuracy. These improvements are slightly smaller than those achieved on the SYSU-MM01 dataset. This difference can be attributed to the larger scale and relatively higher image quality of the SYSU-MM01 dataset, as large-scale, high-quality data can promote model performance.

## 6    CONCLUSION

In the context of VI-ReID, we introduce a novel approach called Cross-Modality Masked Pretraining (CMMP). This method begins by generating images through random channel exchange, creating images that resemble infrared images visually while preserving the texture details of visible images. This step aims to bridge the modality gap between the two types of images. Subsequently, CMMP employs a mask reconstruction paradigm to jointly model visible and infrared information, further enhancing the fusion of these modalities to improve model performance and fine-tuning capabilities.

Through a series of ablation experiments, we identify the optimal mask strategy and prediction targets while determining the best number of pre-training iterations. CMMP not only surpasses the baseline network CAJL Ye et al. (2021) but also outperforms other state-of-the-art methods on two VI-ReID datasets. This showcases CMMP's effectiveness in reducing modality discrepancies and facilitating the fusion of cross-modality information.

AUTHOR CONTRIBUTIONS

Haoyan Ma: Conceptualization, Methodology, Software, Writing – original draft. Xiang Li: Methodology, Validation, Formal analysis, Investigation. Xia Yuan: Visualization, Funding acquisition. Jie Li: Methodology, Writing – review & editing. Chunxia Zhao: Supervision, Project administration.

ACKNOWLEDGMENTS

This work was supported by the National Science Fundation of China under Grant No. 61773210 and the Young Scientists Fund of the National Natural Science Foundation of China under Grant No. 62206134.

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
