# OpenReview forum: "Cross-Modality Masked Pre-training for Visible-Infrared Person Re-identification"
_ICLR.cc/2024/Conference — Submitted to ICLR 2024_

### Official Review · Reviewer_JDzU · 2023-10-26

**Soundness:** 1 poor
**Presentation:** 1 poor
**Contribution:** 1 poor
**Rating:** 1
**Confidence:** 4

**Summary:**

This paper includes AUTHOR CONTRIBUTIONS and ACKNOWLEDGMENTS sections in the main text. Therefore, this paper violates the double-blind reviewing rules.

**Strengths:**

-

**Weaknesses:**

-

**Questions:**

-

**Details Of Ethics Concerns:**

-

---

### Official Review · Reviewer_t1sC · 2023-10-26

**Soundness:** 2 fair
**Presentation:** 2 fair
**Contribution:** 3 good
**Rating:** 5
**Confidence:** 5

**Summary:**

Existing visible-infrared person re-identification(VI-ReID) methods often overlook the disparity between pre-trained data (ImageNet) and VI-ReID data. This paper introduces a novel approach called Cross-Modal Masking Pre-Training (CMMP) for the recognition of visible and infrared humans. Notably, performance enhancements were realized through an initial pre-training stage involving masking and reconstruction, followed by fine-tuning.

**Strengths:**

1. This paper introduces a novel approach to the VI-ReID task, focusing on model pre-training for the first time.
2. This paper has developed a lightweight pre-training method and demonstrated its effectiveness in enhancing VI-ReID.

**Weaknesses:**

While the focus of this paper is innovative, the experimental results indicate a weakness in its performance.

**Questions:**

1. In this paper, the pre-training data and fine-tuning data are identical. If we conduct pre-training based on ImageNet and then proceed with fine-tuning, does this approach offer any performance advantages compared to the optimal method?
2.This paper mentions “When attempting to predict all three cross-modality images simultaneously, the model’s performance is notably worse, considerably lower than when predicting other targets.” Does this observation suggest that pre-training doesn't necessarily require infrared image data, and instead, we could use visible and generated images from ImageNet datasets or large ReID datasets for the pre-training process?

---

### Official Review · Reviewer_HM7x · 2023-10-31

**Soundness:** 2 fair
**Presentation:** 3 good
**Contribution:** 1 poor
**Rating:** 3
**Confidence:** 5

**Summary:**

This paper presents a mask-based pretraining strategy for visible-infrared person re-identification. Based on the extended images via random channel exchange, the proposed CMMP employs a mask-sharing mechanism to simultaneously mask three types of images. Subsequently, the model is encouraged to reconstruct the masked regions via a lightweight decoder.

**Strengths:**

Good writing: I can easily catch the main story of this paper, and the organization of the paper is clear.

**Weaknesses:**

1.	Limited novelty: I can hardly catch the significant contribution and inspiring insights for this field. Since both the model architecture and data generation method in the paper are based on existing methods, thus, what is the core value or unique contribution of this paper?
2.	Lack of citations to recent literature: This paper lacks citations to recent literature, with only one paper from 2021 mentioned in the related work section.
3.	Outdated experimental comparisons: The experimental results lack comparisons to the latest work. Since this is a submission to ICLR 2024, it is inappropriate to rely solely on experimental comparisons with those prior to 2021.
4.	Inaccurate statement about SimMIM: The statement that SimMIM only encodes the unmasked parts is incorrect. SimMIM also encodes the learnable mask features.
5.	Lack of clarity in explanations: The paper repeatedly mentions "misalignment inherent in the VI-ReID dataset" without providing an explanation of the negative impact it causes or whether any existing work has addressed this issue. Additionally, the descriptions of pretraining and finetuning processes are not clear. It is essential to provide clear and detailed explanations to enhance the understanding of the proposed method.
6.	It is not much of a contribution for firstly introducing the mask reconstruction paradigm into VI-ReID. Clarifying the rationality and necessity of introducing it into the task is more important and meaningful.
7.	The and writing and reference format are sometime irregular, such as ‘Ye et al.Ye et al. (2021)’ in Section 2.2, ‘10 experiments’ in Section 4.1 and ‘288*144 size’ in Section 4.2.
8.	For the ablation study on the masking strategy, the experimental comparisons with more masking strategies should be provided, rather than only the random mask.
9.	There is a lack of insight into the experimental results, and the view is superficial analysis on results.

**Questions:**

1.	Both MAE and SimMIM, the most relevant models to this paper, adopt the ViT as the backbones. However, the proposed CMMP model uses a CNN architecture. Could the authors provide the reason for this choice? Did you try to apply CMMP on the ViT?
2.	Overall, the encoding strategy of CMMP seems more similar to SimMIM. Have the authors attempted the MAE-like strategy that only encodes the unmasked parts? Given that CMMP has an 80% mask rate, using the MAE-like encoding could significantly improve training efficiency.
3.	The paper mentions the statement multiple times -- “Due to the misalignment inherent in the VI-ReID dataset, CMMP focuses its reconstruction efforts solely on the visible image and the generated images after masking”. Could the authors explain what this misalignment refers to and why reconstruction is not performed on the infrared image?
4.	Could the authors provide a more detailed explanation of the finetuning process? Specifically, what are the differences between pretraining and finetuning in this paper?
5.	The encoder processing flow should be explained more thoroughly. Figure 2 shows three input images, while Figure 3 only depicts two input images to the encoder. This discrepancy raises concerns about inconsistency. Please clarify this discrepancy and ensure the consistency of the depiction.
6.	It is recommendable to include more figures in the experimental section. After all, compared to the tables, graphical presentation can provide a more intuitive and direct comparison.

---

### Meta-Review · Area_Chair_U6uP · 2023-12-15

**Metareview:**

All three reviewers give consistent and negative comments, and the major issues are: (1) Limited novelty. The basic idea is based on existing methods. (2) Experimental results are insufficient. Moreover, the authors did not support the rebuttal. After reading the comments, the AC cannot recommend to accept this paper and encourage the authors to take the comments into consideration for their future submission.

**Justification For Why Not Higher Score:**

Please see the detailed meta reviews

**Justification For Why Not Lower Score:**

Please see the detailed meta reviews

---

### Decision · Program_Chairs · 2024-01-16

Reject